# Going with the Flow: Modeling the Tumor Microenvironment Using Microfluidic Technology

**DOI:** 10.3390/cancers13236052

**Published:** 2021-12-01

**Authors:** Hongyan Xie, Jackson W. Appelt, Russell W. Jenkins

**Affiliations:** 1Massachusetts General Hospital Cancer Center, Department of Medicine, Massachusetts General Hospital, Harvard Medical School, Boston, MA 02114, USA; HXIE6@mgh.harvard.edu (H.X.); Jackson_Appelt@DFCI.HARVARD.EDU (J.W.A.); 2Laboratory of Systems Pharmacology, Harvard Program in Therapeutic Sciences, Harvard Medical School, Boston, MA 02215, USA; 3Broad Institute of MIT and Harvard, Cambridge, MA 02142, USA

**Keywords:** microfluidics, 3D tumor models, organoids, organotypic culture

## Abstract

**Simple Summary:**

The clinical success of cancer immunotherapy targeting immune checkpoints (e.g., PD-1, CTLA-4) has ushered in a new era of cancer therapeutics aimed at promoting antitumor immunity in hopes of offering durable clinical responses for patients with advanced, metastatic cancer. This success has also reinvigorated interest in developing tumor model systems that recapitulate key features of antitumor immune responses to complement existing in vivo tumor models. Patient-derived tumor models have emerged in recent years to facilitate study of tumor–immune dynamics. Microfluidic technology has enabled development of microphysiologic systems (MPSs) for the evaluation of the tumor microenvironment, which have shown early promise in studying tumor–immune dynamics. Further development of microfluidic-based “tumor-on-a-chip” MPSs to study tumor–immune interactions may overcome several key challenges currently facing tumor immunology.

**Abstract:**

Recent advances in cancer immunotherapy have led a paradigm shift in the treatment of multiple malignancies with renewed focus on the host immune system and tumor–immune dynamics. However, intrinsic and acquired resistance to immunotherapy limits patient benefits and wider application. Investigations into the mechanisms of response and resistance to immunotherapy have demonstrated key tumor-intrinsic and tumor-extrinsic factors. Studying complex interactions with multiple cell types is necessary to understand the mechanisms of response and resistance to cancer therapies. The lack of model systems that faithfully recapitulate key features of the tumor microenvironment (TME) remains a challenge for cancer researchers. Here, we review recent advances in TME models focusing on the use of microfluidic technology to study and model the TME, including the application of microfluidic technologies to study tumor–immune dynamics and response to cancer therapeutics. We also discuss the limitations of current systems and suggest future directions to utilize this technology to its highest potential.

## 1. Introduction

“The field of cancer research has largely been guided by a reductionist focus on cancer cells and the genes within them—a focus that has produced an extraordinary body of knowledge. Looking forward in time, we believe that important new inroads will come from regarding tumors as complex tissues in which mutant cancer cells have conscripted and subverted normal cell types to serve as active collaborators in their neoplastic agenda. The interactions between the genetically altered malignant cells and these supporting coconspirators will prove critical to understanding cancer pathogenesis and to the development of novel, effective therapies”.Hanahan and Weinberg, Cell 2000 [1]

Tumorigenesis is a multistep progress driven by acquired (and, in some instances, inherited) genetic alterations enabling the transformation of normal human cells into tumorigenic and, ultimately, malignant cells. During this process, tumor cells acquire several hallmarks of cancer, including limitless proliferative potential, immortalization, resistance to cell senescence and apoptosis, escape of cell-cycle checkpoints, sustained angiogenesis, and adoption of an invasive and metastatic phenotype, as detailed by Hanahan and Weinberg in 2000 [1] and 2011 [2]. However, tumors are more than cancer cells; they constantly interact with normal cells, such as fibroblasts, inflammatory cells, and cells forming vasculature or responding to infection and injury, to create the “tumor microenvironment” that leads to the acquisition of hallmark callabilities [3]. The immune system has dual roles in cancer pathogenesis and can be co-opted by tumors to promote tumor growth [4], whereas it also has the potential to restrict tumor growth [5], thereby providing long-term, durable disease control.

Over the past two decades, discoveries of genomic characterization in multiple tumor types have not only advanced our basic understanding of tumor initiation and progression, but also greatly influenced cancer management by developing new therapies targeting oncogenic driver mutations, chromosomal rearrangements, or specific pathways affected by genetic lesions [2,6]. For example, identification of the Bcr–Abl fusion in chronic myeloid leukemia (CML) led to the development of imatinib, the first targeted cancer therapeutic, and marked the beginning of the targeted therapy era [7,8]. As with the advance in genome sequencing technology and comprehensive collaborative efforts across many academic medical centers, many driver mutations have been identified and prompted the development of specific inhibitors or antibodies based on these discoveries, including epidermal growth factor receptor (EGFR) inhibitors targeting specific driver mutations in EGFR in lung cancer [9], trastuzumab for Her2-positive breast cancer [10], and BRAF inhibitors for BRAF-mutant melanoma [11], which now have Food and Drug Administration (FDA)-approved small-molecule inhibitors as first-line therapies. Targeted therapies such as PARP inhibitors [12], CDK4/6 inhibitors [13], mTOR inhibitors [14], and VEGF inhibitors [15] have also been used to target specific pathways involved in cancer growth and metastasis. Additionally, targeting lineage-specific, non-oncogene vulnerabilities [16], such as anti-CD20 monoclonal antibodies for some types of lymphoma and leukemia [17] and Bruton’s tyrosine kinase (BTK) inhibitors and PI3Kδ inhibitors in chronic lymphocytic leukemia (CLL) [18,19], has demonstrated clinical activity in certain leukemias and lymphomas.

Despite the promising results of targeted therapies in certain types of cancer, less than 10% of cancer patients have targetable driver mutations [20]. Furthermore, despite initial high clinical response rates, therapeutic resistance remains common [21]. Acquired resistance to treatment with anticancer drugs can be caused by a variety of intra-tumoral factors, such as drug inactivation, drug target alteration, drug efflux, DNA damage repair, insensitivity to drug-induced cell death, epithelial–mesenchymal transition, and tumor cell heterogeneity [21,22]. For example, while suppressing BRAF signaling in BRAF (V600E/K) mutant melanoma cells is effective, multiple resistance mechanisms have been reported to drive MAPK pathway reactivation [23]. A deep understanding of the mechanisms of anticancer drug resistance has implications of how to circumvent this resistance, such as combination therapy [24] and synthetic lethality strategy [25]. In the case of resistance to BRAF inhibitor discussed above, IGF-1R/PI3K signaling was enhanced in resistant melanomas, and combined treatment with IGF-1R/PI3K and MEK inhibitors was tested to induce death of those resistant cells [23]. This gave the rational to explore BRAF/MEK inhibitor (BRAFi/MEKi) combinations, and three such therapies have been approved by FDA [26]. Noteworthily, as an alternative resistance mechanism, cancer cells recruit other cell types (e.g., cancer-associated fibroblast cells, tumor-associated macrophages) to the TME which can promote therapeutic resistance [27,28,29,30,31,32]. Thus, there are multiple tumor-intrinsic and tumor-extrinsic factors that influence response and resistance to molecular targeted therapies. Elucidating the mechanisms of drug resistance is expected to facilitate development of new anticancer strategies and overcome resistance [2].

## 2. Cancer Immunotherapy

Over the past decade, cancer immunotherapy has revolutionized the treatment of cancer with the approval of monoclonal antibodies targeting coinhibitory immune checkpoints, CTLA-4 and PD-1/PD-L1 [26]. Immune checkpoint blockade (ICB) has proven efficacy in several types of cancer [33,34,35]. For example, 40–45% patients with metastatic melanoma respond to single-agent PD1 blockade [36], and 50% benefit from combination immunotherapy [37]. Notably, durable responses have been found in more than 70% of responding patients [36]. Additionally, early-phase clinical trials of several anti-PD-(L)1/BRAFi/MEKi triplet therapy combinations have shown response rates greater than 70% [26].

Despite the great success of ICB in melanoma and other cancers, most patients experience intrinsic or acquired resistance. To overcome immune resistance, numerous clinical trials are already undergoing to evaluate novel immune modulatory agents alone or in combination with anti-PD-1/PD-L1 therapies [35,38,39,40] However, to date, most of these approaches have failed to translate into meaningful clinical benefit compared to standard ICB [41]. Given these challenges, there has been renewed interest in the development of preclinical models to study human tumor immunity to assess cancer immunotherapy combinations effectively and efficiently. In parallel, more sophisticated preclinical tumor models could also help deprioritize ineffective strategies earlier and permit greater focus on more promising approaches. Additionally, establishing such a clinically relevant model is expected to advance our understanding of other tumor-extrinsic components in the TME that affect immune responses, as well as study mechanisms of response and resistance to ICB and nominate targets for next-generation cancer therapeutics [35,38].

## 3. Modeling the Tumor Microenvironment

“All models are wrong, but some are useful”.George E.P. Box (British statistician)

Complex cellular interactions in the TME influence response and resistance to cancer therapies, including ICB [42]. Developing more sophisticated and clinically relevant TME models can not only provide a reliable approach to evaluate the efficacy of novel therapeutic regiments, but also advance our understanding of the interaction between tumor cells and the TME, which in turn will further promote the identification of effective anticancer strategies. Along with the development of bioengineering and animal models, multiple complicated in vivo, 2D, and 3D cancer models have been developed (Table 1).

### 3.1. Tumor Heterogeneity and Composition of the TME

Tumors are formed and developed in the TME, which contains not only tumors cells but also stromal and immune cells [65]. Tumor initiation and progression are influenced by both inherited or acquired mutations within a tumor and the interaction with the multiple components in TME around a tumor, including cells, signaling factors, and supportive structural molecules. Notably, tumor heterogeneity exists between different patients, as well as within different lesions from the same patient, and in different regions of a single tumor. Such heterogeneity is also observed within the immune and stromal elements present in the TME [66]. More importantly, such heterogeneity is dynamic over time [67].

The tumor microenvironment is a highly heterogeneous mix of cellular and noncellular components, including fibroblasts, the extracellular matrix (ECM), mesenchymal stroma/stem cells (MSCs), vasculature, smooth muscle cells, immune cells, nerves, and signaling factors. In the context of TME, the complexity and diversity of TME and its influence on response to therapy have been analyzed along with the development of technologies, such as single-cell RNA sequencing, mass cytometry, and multiparametric imaging [42]. Integrating these moderate/high-resolution TME data can estimate the abundance of tumor-infiltrating immune and stromal cells and reveal the heterogeneity in immunological composition and distribution within tumors [42]. On the basis of immunological composition and status [67,68,69], Binnewies et al. previously described three types of TME, consisting of infiltrated–excluded, infiltrated–inflamed, and infiltrated–tertiary lymphoid structures (TLS) [42]. Infiltrated–excluded TME is characterized by the exclusion of cytotoxic T cells (CTLs) from the tumor core and localization of CTLs to the tumor periphery, in contact with tumor-associated macrophages or ‘stuck’ in fibrotic nests. Infiltrated–inflamed TME is characterized by an abundance of PD-L1 expression on tumor and myeloid cells and highly activated CTLs defined by expression of Granzyme B, IFNγ, and PD-1. Infiltrated–TLS TME contains TLSs, with immunological cell composition like that in lymph nodes, including B cells, dendritic cells, and Treg cells. Clinical data suggested that CTLs, conventional dendritic cells (cDC), natural killer (NK) cells, and Th1 helper T cells play an antitumor role among the infiltrating lymphocytes, while other populations such as immunosuppressive tumor-associated macrophages (TAMs), regulatory T cells (Tregs), and myeloid-derived suppressor cells can dampen the antitumor immune response and play a pro-tumoral role [70]. More recently, by analyzing the transcriptomic information of more than 10,000 cancer patients across 20 different cancers, Bagaev et al. identified four TME subtypes—immune-enriched and fibrotic (IE/F), immune-enriched and nonfibrotic (IE), fibrotic (F), and immune-depleted (D) [71]. The IE/F subtype is immune-inflamed and was characterized by CAF activation and upregulated angiogenesis-associated functional gene expression signatures (Fges). The IE subtype is highly infiltrated and shows increased T-cell activity. Both F and D subtypes lack or have minimal lymphocyte infiltration. The D subtype is similar to the previously described immune-desert type, having the highest percentage of tumor cells; the F subtype shows increased expression level of Fges and number of CAFs. Further investigation found that these TME subtypes are correlated with response to immunotherapy, where patients having a favorable immune microenvironment tend to benefit most from immunotherapy. The patients having the IE TME subtype have the most favorable overall survival (OS) and progression-free survival (PFS), whereas the F subtype shows the worst OS [71].

The generation of antitumor immunity is a cyclic process containing seven steps [72]. Briefly, cancer antigens are released from cell-dead cancer cells and taken by DCs, before being presented in the lymph nodes. This leads to T-cell priming and activation in the lymph nodes. Next, T cells migrate to tumor tissue and infiltrate the tumor. Thereafter, T cells recognize cancer cells and mediate killing. Each step in this cycle can be boosted by immune-stimulatory factors resulting in an enhanced T-cell response, while they can also be inhibited by immune-inhibitory factors leading to immunosuppression. As the networks between immune–immune and immune–tumor interactions become better defined, it will become possible to characterize different classes of TME and determine which cells, molecules, and pathways are critical for enhancing antitumor immunity, and in what tumor contexts. In the meantime, while we are modeling the TME, it is necessary to consider the complexity and heterogeneity of tumor cells and the TME.

### 3.2. In Vivo Models

Murine models can provide an intact innate and adaptive immune system allowing tumors from syngeneic transplantation or genetical engineering to be developed under the influence of the TME [73]. Syngeneic implantable mouse tumor models remain the “gold standard” for studying the TME and evaluating immunotherapies. By inoculating spontaneous, carcinogen-induced, or transgenic tumor cell lines into inbred strains such as C57BL/6, BALB/c, and FVB mice, syngeneic mice develop tumors that interact with the host immune system and become immune-infiltrated [73]. Genetically engineered mouse models (GEMMs) utilize tissue-specific expression of oncogenes and/or tissue-specific deletion of tumor suppressors to drive autochthonous tumor growth, providing a native microenvironment and relevant genetic alterations. However, they may have variability in penetrance and latency, as well as low immunogenicity, because of defined genomic alterations. Despite the limitations of these models, their wide applications have provided crucial insights into tumorigenesis, drug resistance, tumor microenvironment reprograming, and novel combinatory immune therapy evaluation [47,74,75]. Given observed differences between human and murine tumors [76,77], researchers have developed humanized mouse models to reconstruct the human tumor–immune system in immunodeficient host mice with matched human tumor cells and immune cells [73,78]. These humanized mice provide a promising preclinical model to study tumor–immune cell interactions and evaluate the immunotherapeutic response. However, they require autologous immune reconstitution and have a relatively low duration of immune reconstitution, limiting their widespread use. Together, although these models are not perfect, they have led to huge progress in evaluating therapeutic efficacy in implanted human tumor tissue with a partially reconstituted human immune repertoire.

### 3.3. 2D versus 3D Culture

Cell culture is a widely used research model to study cell biology, mechanisms of diseases, and drug sensitivity. Culture of cancer cell lines can be thought of in two broad categories—2D culture and 3D culture. In this section, we describe them individually and compare 2D versus 3D approaches to study tumor intrinsic biology and tumor–immune dynamics, as well as their applications in modeling the tumor microenvironment.

#### 3.3.1. 2D Culture

Traditional 2D culture systems that rely on immortalized cell lines have been widely utilized by cancer biologists to study tumor biology and test anticancer therapies given their low-cost and high-throughput capability, especially studying drugs whose mechanism of action is mostly tumor-specific. However, when isolating cells from a tumor mass and then culturing them in 2D conditions (plastic flasks or dishes), cells lose cell–cell and cell–extracellular environment interactions, which are critical for their physiological functions, such as cell differentiation, gene expression, and response to stimulation [79]. Furthermore, as a result of 2D culturing, the cell morphology and cell polarity change [80]. Another shortcoming of 2D culture systems is the lack of complexity and heterogeneity of the tumor microenvironment, making it a less reliable model system to study complex cancer biology and test preclinical drugs [81]. In particular, to further improve the efficacy of cancer immunotherapy, there is an increasing need to use methods to model the TME to further drive immunotherapeutic drug development and rational combination immunotherapy [82,83,84]. Because of the disadvantages of 2D systems, there are some “2D culture variant systems” that have been demonstrated. For example, several groups have described the method of keeping cell polarity in 2D culture systems by culturing cancer cells on a model ECM substrate [85,86] or via custom micropatterned substrates such as micropillars [87].

#### 3.3.2. 3D Culture

Three-dimensional (3D) tumor models are increasingly applied to more faithfully recapitulate biology observed in vivo [88]. Central to the development of 3D tumor models are both 3D culture devices and biological materials (“biomaterials”) to serve as the scaffold in which tumor cells, spheroids, or organoids live and grow. While biomaterials can be natural or artificially made [89], most 3D tumor models use natural biomaterials purified from animals or plants that can be enzymatically digested by tumor and/or stromal cells. The selection of the biomaterial to be used as a scaffold for 3D tumor modeling is far from trivial as some are biologically active (e.g., ECM components driving cancer growth [89]) and can offer specific advantages, such as sustaining drug release or generating bone-like structures [90,91]. Commonly used natural biomaterials in 3D tumor models include collagen, gelatin, Matrigel, hydrogel, chitosan, alginate, silk, poly-ε-caprolactone (PCL), and hyaluronic acid [89,92,93]. These biomaterials can be used in alone or in combination and applied to culture cells or spheroids. Beyond that, some studies have also reported engineering biomaterials to have specific functions. For example, gelatin hydrogel microspheres (GM) have been used as a source to continually release drugs [90,94]; PCL scaffolds have been engineered to have bone-like architecture and mineralization [91,95].

By integrating 3D culture systems and biomaterials, multiple 3D culture models, including spheroids, organoids, and 3D bioprinting, have been developed [81,96,97] and applied in various types of cancers, such as melanoma [59,98], breast [99,100,101], prostate [102], bladder [103], pancreas [104], and head and neck [105]. Multicellular tumor spheroids (MTSs) comprise multiple cancer cells that self-assemble into 3D spherical structures. Thus, they can capture cellular interactions in a 3D context and can maintain certain cell morphology, as well as mimic metabolic and proliferation gradients similar to in vivo conditions. MTSs are usually generated by culturing cells in low-attachment plates, hanging droplets, and scaffolds [96,106]. Depending on the cell sources they are created from, such as cell lines, multicellular mixtures, and patient-derived tissues, different models have been demonstrated [107]. Cancer cell line-derived MTSs are commonly used for convenience and ease of generation, as well as for variety of application, including drug screening and evaluation of drug penetrance [97]. As cancer cell line-derived MTSs lack immune cells, a variety of coculture systems have been developed (discussed further in Section 3.3.5). Together, with more physiological and clinical relevance, there is increasing interest in tissue-related spheroids for providing a robust approach in pursuing precision and patient-specific therapy.

Organoids are 3D cell clusters generally created from stem or progenitor cells and embedded in an extracellular matrix to spontaneously form organ-like or tissue-like structures with cell types typically present in original tissue [85,108,109]. They can be expanded for long-term culture, preserve histological and genetic features of the native tissue of origin, and are amenable to cryopreservation to facilitate long-term, iterative experimentation. As organoids are also amenable to genetic manipulation, organoid-based 3D tumor models have been widely applied in many areas, including cancer research [110]. Currently, organoids models have been generated using a variety of different normal and tumor tissues, such as lung [111], breast [112], colon [113], and liver [114]. Notably, organoids generated from tumor specimens grown in air–liquid interface (ALI) culture systems can preserve the stromal and immune components of the TME [107]. These patient-derived organoids (PDO) open new avenues for studying tumor–immune dynamics and cancer therapeutics. They also hold promise for wider application in combination with other technologies, such as gene editing and microfluidic devices. However, it is worth considering the impact of organoid media on cell differentiation, as well as some other factors that will affect drug response, as it has been reported that PDO-based drug efficacy evaluations fail to present the same response in the clinic [115].

3D bioprinting is an innovative computer-aided engineering technique to generate organized 3D tissue structures comprising multiple cell types in physiologic 3D models. It can effectively recapitulate key components of the TME and is amenable to high-throughput screening or testing drug efficacy in various cancer models [116,117]. However, 3D bioprinting studies are currently limited by challenges studying autologous immune and stromal cells mixed with tumor cells in favor of cell-line based reconstitution studies in which transformed immune and tumor cells are cocultured.

#### 3.3.3. Comparison between 2D and 3D

3D cell cultures differ from traditional 2D cultures in terms of cell features, cell–cell interactions, cellular mechanics, and nutrient gradients. Depending on the type of culture chosen, cell behavior differs in many aspects, as reviewed elsewhere [118]. Here, we discuss cell growth, differentiation, gene expression, and drug sensitivity.

Several studies have compared the effects of 2D and 3D culture methods on cell growth, gene expression, and differentiation. Chitcholtan et al. showed that tumor cells lines in 2D culture have a higher proliferation rate compared to 3D culture with a reconstituted basement membrane (rBM). However, cells in 3D cultures increase the expression of β4 and β1 integrins, indicating enhanced polarization and differentiation. Lee et al. found that spheroids generated from oral cancer cell lines have a high proportion of cancer-initiating cells, which is probably due to 3D culture-induced EMT with the support of the downregulation of E-cadherin and upregulation of fibronectin, Sox2, Oct4, and Nanog. Moreover, CD133 and ALDH, two putative stem-cell markers were observed to be increased in 3D culture conditions [119].

Several studies have also reported that tumor cells are less sensitive to anticancer drug treatments in 3D compared to 2D culture. For example, a recent study reported that aggregated spheroids derived from breast cancer cell lines can prevent paclitaxel-caused apoptosis [120]. Similar findings were observed with prostate cancer lines exposed to antineoplastic drugs paclitaxel and docetaxel in 2D and 3D culture conditions [121]. Another separate study reported that an ex vivo 3D Ewing sarcoma model cultured in electrospun PCL scaffolds was more resistant to drug treatments in comparation with 2D culture and had significant differential gene expression enriched in insulin-like growth factor-1 receptor and rapamycin pathways [95]. Given that cells cultured in 3D have a gradient from the surface to the center of a spheroid or organoid in terms of attaining ingredients from the cultural media, whereas cells in 2D have equal access to nutrients and drug, cells, especially in the center of the 3D setting, tend to be less sensitive to drugs due to this gradient. Notably, this difference in implied geometry between 2D and 3D cultures is an important factor to understand physiological response [118].

In this regard, 3D culture offers a more physiologically relevant environment to evaluate drug efficacy, as well as an ideal system to screen potential drug targets. For example, Takahashi et al. performed 3D culture-based CRISPR to identify NRF2 as a target to induce ferroptosis death in lung tumor spheroids cells [122]. However, it must be emphasized that the platform of choice is often dictated by the specific process of interest, and that model-specific features may influence the findings, necessitating cross-model validation.

#### 3.3.4. Evaluating Tumor–Immune Interactions in 2D Culture Systems

Given the complexity of tumor–immune cell interactions in vivo, several commonly used reductionist systems have been reported to study these interactions via 2D coculture system. Coculture systems with defined TCR-specific T cells recognizing well-characterized antigens are powerful tools, such as the murine transgenic OT-1 mice designed to recognize the SIINFEKL peptide derived from the xenoantigen ovalbumin (OVA) expressed by tumor cells [55]. For in vitro studies, CD8 T cells isolated from spleens or lymph nodes from OT-1 T-cell receptor (TCR) transgenic mice are cocultured with syngeneic tumor cells stably expressing OVA antigen. Since OT-1 cells have a TCR that is specific for OVA, this system has been widely used to study T-cell biology, tumor sensitivity, and screening targets of sensitizing T cell-mediated attack [55]. Additionally, a variant coculture system has been reported by several groups via in vitro transduction of a TCR into primary mice or human T cells and paired antigens to tumor compartments [54,57,123]. Other studies have described coculture systems using in vitro CD8 expanded T cells with matched tumor cells from both mice and humans. For example, Gestermann et al. demonstrated a human autologous melanoma–T cell coculture and found that LAG3 and PD1 plus LAG3 inhibition can promote antitumor immunity [56].

#### 3.3.5. Evaluating Tumor–Immune Interactions in 3D Culture Systems

With the increasing development of tissue culture technology, 3D culture models have been used to advance many fields of cancer immunology research. Given the homogeneity of cell line-based spheroids, researchers have tried to coculture spheroids and immune components derived from matched tumors to study tumor–immune interactions [124]. Studies has been reported to coculture tumor spheroids with T cells [124], macrophages [105], stromal fibroblasts [125], and NK cells [126]. As the study of autologous immune cells is more biologically relevant and may offer greater insights compared to 3D models reliant on a heterotypic culture of unrelated cell lines, evaluation of patient-derived tumor spheroids with autologous immune cell types has gained interest. It has been reported that NK and T cells are able to infiltrate into colorectal tumor spheroids and affect their viabilities, accompanying the upregulation of HLA-E, an inhibitor ligand of NKG2A expressed by NK and CD8 T cells [127]. Recently, Blasio et al. reported a human organotypic skin melanoma culture (OMC) system, leveraging the decellularized dermis as a scaffold to coculture keratinocytes, fibroblasts, and immune cells with melanoma cells. By reconstructing the TME with multiple cell components, tumor growth was observed; supportively, the characterization of the conversion of cDC2s into CD14^+^ DCs indicated an in immunosuppressive phenotype [128]. Another group utilized a coculture system of monocytes with autologous spheroids from head and neck squamous cell carcinoma (HNSCC) or its benign control to predict prognosis by analyzing the coculture secretion [129]. They found that secreted IL-6 in a coculture of monocytes and benign spheroids can predict recurrence and prognosis, whereas, in a coculture with monocytes and malignant spheroids, it predicts recurrence; on the other hand, another secretion, monocyte chemoattractant protein (MCP)-1, did not predict prognosis. Moreover, 3D spheroid culture can be integrated with other technologies to extend its application. For example, to identify potential targets to enhance antibody-dependent cell-mediated cytotoxicity (ADCC), researchers incorporated a 3D tumor spheroid microarray in the development of a high-throughput screening system to study natural killer cell-mediated cytotoxicity [130]. Briefly, through 3D coculture of NK92-CD16 cells with pancreatic (MiaPaCa-2) and breast cancer cell lines (MCF-7 and MDA-MB-231) in a 330 micropillar–microwell sandwich platform, cancer cells showed a dose response to paclitaxel and antibodies. An additional approach of using spheroids to advance cancer immunotherapy has been demonstrated by incorporation spheroids with microfluidic devices, as described in more detail in Section 5.

Although the lack of immune components in epithelial-only PDOs limits their application in functionally modeling the ICB response, multiple reports have demonstrated the potential use of organoid technology in the study of cancer immunotherapy, either by coculturing organoids with immune cells or using the ALI method. Dijkstra et al. reported that cocultures of autologous tumor organoids and paired peripheral blood lymphocytes can be leveraged to enrich tumor-reactive T cells [62]. Further analysis indicated that these T cells can be further used to evaluate the efficiency of killing of matched tumor organoids, providing an approach to assess the sensitivity of tumor cells in response to T-cell-mediated killing. Another study demonstrated the construction of complex organotypic models via coculture of primary pancreatic cancer organoids with stromal and immune components of the tumor microenvironment, which in this context can successfully induce cancer-associated fibroblast activation and tumor-dependent lymphocyte infiltration [104]. Additionally, several similar organoid models generated from melanoma, chondroma, glioblastoma, and colorectal carcinoma have been developed to study responses to immune checkpoint blockade via coculture systems [63,131,132,133]. In another aspect, Neal et al. validated ALI as a method of propagating PDOs or mouse tumors in syngeneic immunocompetent hosts to preserve immune cells (T cells, B cells, NK, and macrophages) infiltrated in tissue and the TCR spectrum. They further demonstrated that human and murine PDOs can be successfully used to model the response to ICB by observing the activation of tumor antigen-specific TILs and tumor killing [64].

## 4. Microfluidic Technology

Microfluidics is a technology that allows one to manipulate tiny (10^−9^ to 10^−18^ L) amounts of fluids to flow in channels of hundreds of millimeters in size [134,135,136,137]. As the cell volume-to-extracellular fluid volume ratio is more than one for tumor cells and immune cells within the TME (i.e., smaller volumes of extracellular fluid), the size scale of microfluidic devices makes them very suitable for biological application to study and model the TME [135]. Another feature of microfluidic technology is its low Reynolds number (Re), which refers to the ratio of inertial to viscous force on a fluid [135]. With a low Re value, the fluid flow in microfluidic systems is laminar, meaning that mass is transferred mainly through diffusion. This makes it possible to generate a concentration gradient of soluble factors temporally and spatially in a microfluidic system. Microfluidic devices are also called “organ chips” or “tissue chips”, using plastic material or other optically clear materials to form perfused hollow microchannels, which can mimic vasculature [83,138]. While polydimethylsiloxane (PDMS) is often used to make microfluidic devices, several prototypes of rigid thermoplastic polymers (e.g., polycarbonate, cyclic olefin copolymer) have been validated to overcome the key limitations of PDMS, such as the adsorption of hydrophobic molecules and evaporation [139,140,141]. Additionally, as with the development of microfluidic technology and the enthusiasm to optimize this system to better mimic physiological conditions, many complex microfluidic devices have been developed for specific functions [142,143,144,145,146,147]. Overall, in the past decade, microfluidic technology has been greatly developed, and numerous advantages have extended their applications in a variety of fields, as discussed in detail in the next section.

## 5. Modeling Cancer in Microfluidic Chips

“A key consideration in the development of new microfluidic methods in academic research should be whether the use of microfluidics introduces truly enabling functionality compared to current methods. When a potential application passes this test, the chances of contributing useful technology to the field are substantially higher”.Sackmann et al. Nature 2014 [134]

By integrating microfluidic technology with 3D culture systems, researchers can control matrix structure, matrix stiffness, cellular composition and ratio, flow rates, and other features. Those devices can also be combined with or applied to high-resolution and real-time imaging to explore various preclinical analysis in a specific organ or disease context [83,138]. Numerous kinds of microfluidic devices have been developed to capture structural and functional properties of human organs or organ-specific disease states. Because of these advantages, microfluidic devices are increasingly used not only to study cellular processes essential for cancer growth and progression, but also for preclinical drug testing using clinically relevant physiologic conditions [116]. As such, microfluidic device-based cancer models may pave the way for developing platforms for functional precision cancer medicine to perform drug sensitivity testing using “living biopsies” from cancer patients to inform therapeutic decisions. Microfluidic technology has been applied to the study of numerous aspects of cancer biology and cancer immunotherapy (Figure 1 and Table 2), including tumor growth, cancer cell extravasation, angiogenesis, immunotherapeutic response, and drug screening, as discussed below.

### 5.1. Tumor Growth

One of the hallmark features of cancer cells is sustained proliferative signaling [2], leading to uncontrolled cell growth. 3D microfluidic devices have been used to explore interactions among different cell subtypes (e.g., cancer-associated fibroblasts, CAFs) and factors (e.g., extracellular pH) in the TME that affect cancer cell proliferation. To investigate the effect of fibroblasts and ECM proteins on cancer cell growth and migration, Lugo-Cintrón et al. cocultured breast cancer cells with fibroblasts in a microfluidic device. They found that fibroblasts promoted cancer cell growth and induced more migration by increasing the level of metalloproteinases (MMPs) in media [99]. To investigate the effect of pH on tumor viability, Lam et al. employed a bifurcated microfluidic device and compared cell proliferation capability between culturing MDA-MB-231 breast cancer cells in media with and without fibrin, which can interact with acid-neutralizing calcium carbonate (CaCO_3_) nanoparticles [148]. The authors found that nanoCaCO_3_ treatment inhibited tumor cell growth, whereby the media pH was increased from 7.14 to 7.25 and the intracellular pH was decreased from 7.6 to 7.05, suggesting that low pH promotes cell proliferation. Using the same device, they also cocultured cancer cells and fibroblasts, followed by treatment with nanoCaCO_3_, further confirming that nacoCaCO_3_ can specifically inhibit the growth of tumor cells rather than surrounding fibroblasts. The nanoCaCO_3_ treatment buffered the pH within the normal physiological range and inhibited tumor cell proliferation. Such a model allows studying tumor growth in a 3D culture environment, as well as the effect of pH on tumor growth, which a 2D system cannot provide.

### 5.2. Tumor Migration and Extravasation

Cancer cell extravasation is the process whereby circulating tumor cells transmigrate through blood vessels to form deposits at secondary sites. Microfluidic modeling has enabled evaluation of both tumor cell/endothelial cell migration and extravasation of tumor cells from model endothelial-lined blood vessels. Compared to conventional methods of studying cell migration (e.g., transwell assays and scratch assays), such a microfluidic-based platform uses a monolayer of endothelial cells in the media channel to mimic microvasculature, providing a more physiologically relevant microenvironment to investigate cancer–vascular crosstalk and to identify factors involved in the regulation of the migratory potential of tumor cells. By utilizing this technology, several secreted factors have been reported to be involved in cancer extravasation by regulating cancer–vascular crosstalk [149]. In particular, the authors established an organotypic microfluidic model by coculturing breast cancer cells and endothelial cells derived from pluripotent stem cells in a collagen–fibrinogen matrix and found that increased levels of secretion of IL-6, IL-8, and MMP-3 were positively correlated with extravasation [149]. In a separate study, Chen et al. developed a microfluidic device with a central gel region suspended with human umbilical vein endothelial cells (HUVECs) and two side gel regions suspended with normal human lung fibroblasts (NHLFs), with each gel region flanked by two media channels. Using such a microfluidic device, the authors demonstrated that β1 integrin was required for tumor cell to form stable protrusions and initiate migration [151]. A different study from the same group, using microfluidic devices having a vasculature compartment formed by HUVECs in fibrin gels, Chen et al. demonstrated enhanced melanoma cell migration promoted by tumor cell-derived CXCL1 and by neutrophil-derived IL-8 [150]. Tumor-associated macrophages (TAMs) have been shown to promote tumor cell migration. When inflammatory breast cancer cells were treated with macrophage-conditioned medium, they became more migratory. Further analysis showed that several macrophage-derived chemoattractants (e.g., interleukins 6, 8, and 10) contributed to this metastatic phenotype [187]. Underscoring the differences between experimental systems, a similar study demonstrated a role for TAM-derived TNFα in promoting the migratory capability of tumor cells [100].

### 5.3. Angiogenesis

Angiogenesis is of particular importance for tumor growth. Multiple microfluidic-based angiogenesis models have been developed to mimic the initiation of new vessel formation in vitro [152]. One of these studies demonstrated reconstituting angiogenic sprouting in a microfluidic device and how to apply this model to identify the effect of potential angiogenic inhibitors on sprouting morphogenesis in vitro [152]. In particular, using this model, the authors observed a series of key events of neo-vessel formation within a 3D extracellular matrix. The authors further explored the role of angiogenesis inhibitors, such as vascular endothelial growth factor (VEGF) receptor 2 (VEGFR2) inhibitor and sphingosine-1-phosphate receptor (S1PR) inhibitor, trying to connect a specific stimulus to a defined morphogenetic process. In addition, there is a model trying to integrate mathematical and computational methods within a microfluidic platform. This combination represents a powerful way to test multiple experimental parameters on cell migration and angiogenesis. Ayensa-Jimenez et al. utilized this approach to examine how cells responded to VEGF gradients and how cell migration was affected by cell density and by the device features such as width and length [153].

### 5.4. Cancer Metastasis

Metastasis is a complex process, during which a few cells from the primary tumor migrate to a secondary organ after going through a series of sequential steps [188]. Metastasis contributes to 90% of human cancer deaths [189]; thus, it is of importance to investigate the mechanisms underlying this critical step in cancer progression. Emerging data have reported multiple factors to be implicated in metastasis. For example, Cho et al. [154] developed a three-channel microfluidic device having a lymph vessel–tissue–blood vessel structure to study the effects of inflammatory cytokines in lymphatic metastasis and found that IL-6 induces epithelial–mesenchymal transition (EMT) by mediating intercellular interactions in the TME. A similar study utilized a two-channel system—one channel coated with endothelial cells to mimic vasculature and the other containing breast cancer cells embedded in Matrigel—and found that proinflammatory cytokine TNFα is necessary for cancer metastasis [155]. In addition to secreted factors, tumor cell integrins, and ECM components, immune cells have been reported to promote or inhibit cancer metastasis. Kim et al. described the functions of macrophages and monocytes in regulating the formation of the cancer metastatic niche, identifying a novel role for monocyte-derived matrix metalloproteinase 9 in cancer cell extravasation [190].

### 5.5. Modeling the TME

Tumor models that faithfully recapitulate key components of the TME may facilitate applications to predict sensitivity to cancer therapeutics, paving the way for personalized or precision functional medicine strategies. Microfluidic devices are ideal for such applications, as multiple components in TME can be individually controlled. For instance, Jeong et al. demonstrated a microfluidic chip allowing the coculture of tumor spheroids with CAFs to monitor their reciprocal interaction, providing a platform to study the crosstalk of tumor cells and CAFs [156]. Another similar approach has been adapted to model tumor cell–stroma interactions [157], as well as the interaction between tumor and endothelial cells [158]. Additionally, microfluidic devices have also been developed to mimic other features of the TME. For example, Michna et al. described a new platform containing interconnected microchannels to model a highly vascularized system [159]. Huang et al. discussed the approaches of modeling biophysical features, such as the ECM and interstitial flow in TME [160], while other studies focused on developing microfluidic chips to control oxygen concentration [161,162]. In another aspect, 3D culture technology (e.g., spheroid, organoid) allows keeping the naïve TME components on devices, providing more physiological relevance. One such approach is based on murine- and patient-derived organotypic tumor spheroids (MDOTSs/PDOTSs), of which the cellular compositions include not only tumors cell, but also lymphoid and myeloid populations and subpopulations, making it an ideal system to study the TME ex vivo [60,138]. Furthermore, MDOTSs/PDOTSs have also been used to evaluate therapeutic response, as described in Section 5.8 and elsewhere [60].

### 5.6. Immune Cell Migration/Recruitment

Modeling immune cell migration is vital to understand tumor–immune dynamic interactions and immunotherapeutic response. By integrating microscopy technology, cell migration modeled by microfluidic chips can be monitored in real time. Numerous studies have employed microfluidic technology to explore the role of immune cell migration in cancer development, as well as identify the potential factors affecting this process and the effect of migration on the response to immunotherapy. One study demonstrated the application of a microfluidic system containing lumen-based vascular component to study neutrophil–endothelial interactions, revealing a key role of interleukin-8 (IL-8) in promoting neutrophil chemotaxis and priming [163]. Recruited neutrophils increased the local production of reactive oxygen species (ROS), which promoted increased cell adhesion and upregulation of chemokine receptors.

On the other hand, cancer cells also play an important role in affecting immune cell migration, which is mostly through secretion of cytokines or chemokines. By coculturing human pancreatic adenocarcinoma cells and macrophages in a microfluidic device, one group found that those tumor cells promote macrophage migration by secreting chemokines IL-8 and C–C chemokine ligand 2 (CCL2). A similar study demonstrated that bladder cancer cells inhibit antitumoral M1 macrophage polarization but promote pro-tumoral M2 macrophage polarization through lactate-mediated macrophage chemotaxis [103]. Another study elucidated that chitinase 3-like 1 (CHI3L1), an enzymatically inactive mammalian chitinase, interacts with the extracellular matrix of melanoma cells, increasing the secretion of various cytokines, such as CCL2, and growth factors, such as vascular endothelial growth factor A (VEGF-A) [98]. Therefore, the secretome influences immune cell recruitment to the vascular endothelium, in turn affecting immunotherapeutic response.

### 5.7. T Lymphocyte Activation

An effective response to ICB involves effector CD8^+^ T function. Although multiple assays have been employed to detect T-cell activation and effector T-cell function, several studies have reported different levels of T-cell activation between 2D and 3D models [191,192]. Thus, it is valuable to monitor T-cell activation within 3D environments in real time under both native and stimulus conditions. Kirschbaum et al. described an approach to activate T cells mediated by contact with anti-CD3/anti-CD28-presenting microbeads, using a chip containing microelectrodes for dielectrophoretic manipulation; this allowed the assembly of specific beads on cells, which could further induce T cells to express CD69 after overnight cultivation [165]. Park et al. employed a microfluidic system to monitor the interaction between leukocytes and endothelial cells and applied it to identify potential drugs that may modulate this interaction [166]. By controlling shear stress, the authors developed a microfluidic environment where activated T cells were able to bind to HUVECs pretreated with tumor necrosis factor-alpha (TNF-α). To confirm whether this system could monitor T-cell activation, the authors cocultured autoreactive T cells from patients with systemic lupus erythematosus (SLE) and activated HUVECs and found higher binding ability in comparison with incubated with T cells from a person without SLE. Next, the authors further investigated the role of immunosuppressors tacrolimus and cyclosporin A in blocking the bindings of these autoreactive T cells to HUVECs.

### 5.8. Therapy Assessment

Despite the great success of cancer immunotherapy in recent years, long-term durable responses are still observed in a minority of patients due to intrinsic and acquired resistance. Rational combination therapy has shown promising results in many types of cancer compared with monotherapy. As the number of combination therapies is ever increasing, there is an unmet need to develop a platform that can accurately predict the efficacy of these therapies in the preclinical and clinical settings in a timely manner. 3D microfluidics has been used to test immunotherapeutic response to ICB alone or in combination with other regimen(s) using MDOTSs/PDOTSs [60,138]. To leverage this system to evaluate immunotherapeutic response, a variety of sensitive and resistant syngeneic models were used to validate the robust functions of this system. Using fluorescence-based live/dead imaging, the authors demonstrated T-cell-mediated killing ‘on chip’. Then, such a strategy applied to MDOTSs/PDOTSs, revealing that TBK1/IKKε inhibition can sensitize melanoma cells to PD-1 blockade [60]. Using same approach, Deng et al. identified that a combination of CDK4/6 inhibitors and PD-1 blockade enhances treatment efficacy by promoting T-cell infiltration and activation [58]. Meanwhile, Sade-Feldman et al. performed scRNA-seq and defined two different CD8^+^ T cell states, CD8^+^CD39^−^TIM3^−^ (DN, double-negative) and CD8^+^CD39^+^TIM3^+^ (DP, double-positive), which can be used as a predictor for the success or failure of checkpoint immunotherapy [59]. To further validate this, DN and DP CD8^+^ T cells were enriched from CT26-GFP murine tumors and reintroduced into MDOTSs with 100-fold DN CD8^+^ T cells, DP CD8^+^ T cells, or a 1:1 mixture (DN:DP) with or without anti-PD1 treatment. The authors found that DN cells support antitumor activity in this ex vivo system. Other forms of cancer immunotherapy include cell-based immunotherapy and cytokine therapy. One study developed a microfluidic model with one monolayer of endothelial cells adjacent to breast cancer spheroids on each side and showed enhanced cytotoxicity around spheroids when treated with natural killer cells and a combination of antibody-cytokine regimens [167]. Another study leveraged spheroids containing cancer cells and fibroblasts with the addition of PBMCs to test the efficacy of a new immune cytokine and T cell bispecific antibody with or without IL-2 [193].

### 5.9. Disease and Therapeutic Response Monitoring

Disease and therapeutic response monitoring is required for making clinical decisions and prognostication. Numerous microfluidic devices have been adapted to function in these areas. Currently, a microfluidic device designed by CellSearch has been approved by the FDA to predict prognosis and evaluate progression-free survival and overall survival of patients [168]. This device exploits antibody-coated magnetic particles targeting EpCAM to detect and quantify circulating tumor cells (CTCs) of epithelial origin in the whole blood of patients with metastatic breast cancer, prostate cancer, and colorectal cancer [168,194]. By setting a predetermined threshold, the number of detected CTCs is used as a parameter to predict prognostic outcome. Beyond this, there are several other devices that can be used for CTC isolation [169,170,171,172]. In addition to CTCs, several immune-affinity microfluidic devices have been applied to capture exosomes in liquid biopsy. One of them is from ExoSearch. With enriched blood plasma exosome in microfluidic devices, immunomagnetic beads are employed to capture and measure exosomal tumor markers (such as CA-125, EpCAM, and CD24) [173]. A similar device from ExoChip uses antibodies against CD63 to isolate, quantify, and recover exosomes with intact RNA for exosomal microRNA profiling via open array analysis [174]. Another device called the nPLEX (nano-plasmonic exosome) sensor simultaneously isolates the exosome through CD24, CD63, and EpCAM markers and detects them by surface plasmon resonance (SPR) [175]. As some exosomes have been shown to dampen antitumor immunity, it is promising to monitor immunotherapeutic response-related markers (e.g., PD-L1) on the exosome to predict ICB efficacy [195].

Single-cell RNA sequencing has proven to be a powerful and transformative technology to study intra-tumoral heterogeneity. Progress in microfluidics technology and development in cellular barcoding have enabled the integration of microfluidic and scRNA-seq, which enables profiling of intra-tumor heterogeneity and deepens the understanding of transcriptional programs and cell states associated with therapeutic evasion. Several groups have reported this application. Demaree et al. reported a single-cell sequencing (SiC-seq) platform with high-throughput and low-deviation characteristics in droplet microfluidics [176]. Han et al. demonstrated a Microwell-seq platform with high throughput [177]. Habib et al. developed a droplet microfluidic platform (DroNc-seq) to conduct single-cell nuclear RNA-seq [178]. Such an integration of microfluidic technology and scRNA-seq holds promise to further advance clinical validation and develop more effective personalized medicine. To monitor immune cell heterogeneity, Merouane et al. developed a timelapse imaging microscopy-based microfluidic platform in nanowell grids to study cell-to-cell interactions between tumor and immune cells in real time [179]. Another microfluidic platform developed by the Heath group is based on a single-cell barcoding chip (SCBC), containing arrays of microwells with immobilized barcode-like antibodies for proteins and other detections [180]. A similar technique called beads-on-barcode antibody microarray (BOBarray) was reported by Yang et al. [181], whereas Armbrecht et al. proposed single-cell protein profiling with barcode beads [182].

### 5.10. Drug Screening

There has been increased interest in developing 3D tumor models to improve cancer drug development given the limited fraction of candidates that are ultimately FDA-approved and the staggering cost of drug development [196,197,198]. Microfluidic-based drug testing enables a reduction in the volume of reagents required and can be adapted for parallelization and potential automation. Although most microfluidic device-based drug screening applications are at a proof-of concept stage, several groups have tried to develop microfluidic systems to perform drug screening. For example, Bhise et al. developed a liver-on-a-chip platform via 3D printing technology using hepatic spheroids as the material source to directly print liver tissue into the microfluidic device [183]. The authors further proved that bio-printed hepatic spheroids can be cultured long-term in devices and used as a drug toxicity testing platform. In a separate study, Riley et al. described a microfluidic platform that can maintain thyroid tissue slices ex vivo for a minimum of 4 days and be used to evaluate the response of thyroid tissue to radioiodine sensitivity/adjuvant therapies in real time [184]. Schuster et al. reported an automated microfluidic device designed for combinatorial and dynamic drug screening using pancreatic tumor organoids [185]. This microfluidic platform includes 200 individual chambers, enabling the loading of temperature-sensitive gels and an overlaying channel layer, thus allowing to test 20 independent fluidic conditions, with culturing for more than 14 days. Additionally, Pandya et al. demonstrated a microfluidic device developed for drug screening in a 3D cancer microenvironment [186]. In this model, involving the integration of microfluidics and electrical sensing modality, devices can be used for chemotherapeutic drug testing and efficacy evaluations in less than 12 h.

## 6. Challenges, Opportunities, and Future Directions

*Challenges*—Microfluidic 3D MPSs offer several advantages over traditional 2D culture systems for preclinical studies evaluating the TME, although several challenges remain related to pre-device, on-device, and post-device processes and analytics. When considering the use of patient-derived biospecimens, the tumor size and types of biopsies are critical factors that affect the successful establishment of a reliable environment retaining physiologically relevant components of the tumor and TME. For example, core needle biopsies offer scant cellular material; such biopsies may be better suited for organoid culture [199] and/or single-cell RNA sequencing than organotypic cultures. Even with good-quality specimens to study, 3D culture is a time-consuming process requiring several hours to a few days to process specimens, generate cultures, and monitor response. Furthermore, variable parameters in culture conditions, such as media, different ECM components and their concentration, and growth factors can affect the function of whole system. In a preprint study, Raghavan et al. (https://www.biorxiv.org/content/10.1101/2020.08.25.256214v2, last accessed 29 November 2021) compared the transcriptional state of 23 metastatic PDAC needle biopsies and matched 48 organoid models via scRNA-seq. The authors found that the supplement in organoid media can affect transcriptional cell state. In a separate study, Dijkstra et al. found that the murine basement membrane matrix (Geltrex) used in their coculture system can activate human CD4^+^ T cells and prompt nonspecific immune responses [62]. Additionally, some microfluidic device-based applications require expensive microscopes to acquire high-content imaging and deconvolution for 3D imaging [200], limiting their wide application. While reduced materials are needed in microfluidics, the limited number of cells also precludes certain analyses (e.g., Western blotting, mass spectrometry). Beyond this, the isolation and collection of single cells from microfluidic devices are challenging albeit possible.

Although microfluidic technology has been leveraged to model tumor development, tumor–TME interactions, and response to cancer therapeutics with promising and exciting results, the extent to which current microfluidic cancer models approximate biological processes observed in vivo remains unclear. Most microfluidic cancer models are capable of recapitulating specific aspects of cancer biology and/or the tumor–immunity cycle, and it remains to be seen if a single model will be able to mimic the complexity of a living organism. In the short term, it will be important to perform comprehensive analyses of emerging microfluidic-based 3D cancer models to demonstrate the ways in which key biological processes differ in comparison with traditional 2D models. Furthermore, researchers entering the field should be aware of the advantages and limitations of specific model systems in order to select the appropriate 3D model system to study specific biological processes of interest. The importance of cross-model validation cannot be overstated, and key biological insights derived from 3D microfluidic studies should be validated with complementary model systems to ensure scientific rigor.

*Opportunities*—The unique features of microfluidic devices provide many advantages compared with other systems in various aspects. First, the small size of operating systems is compatible with limited input material, making it compatible with samples with limited size, such as patient-derived samples. This facilitates reduced consumption of precious patient materials and reagents, all in an experimental system that offers greater physiological relevance. Microfluidic systems can be used across cancer types to study a variety of important biological processes including tumor–immune dynamics. In the meantime, 3D microfluidic modeling of the TME has demonstrated utility in evaluating novel cancer immunotherapy combinations [58,169,170]. Integration with other established and emerging technologies may facilitate deeper understanding of these model systems. For example, microfluidic-based MDOTS/PDOTS studies have used several terminal assays, including fluorescence-based live/dead assay, bulk RNA sequencing, single-cell RNA sequencing, flow cytometry analysis, and immunofluorescence [60,138]. Several groups [173,174,175] have leveraged microfluidic approaches to enrich and conduct exosome analysis, which can not only increase the sensitivity of assays, but also provide flexibility in design. With advancing developments in tissue engineering, microfluidic systems allow perfusion of vascularized structures to increase microenvironment control and facilitate real-time imaging analysis [201]. Moreover, microfluidic techniques have also enabled efforts to profile the TME [177], identify prognostic biomarkers [169,171], and study the response to ICB [60], thus promoting the development of personalized medicine. Because of these above-mentioned advantages, microfluidics has become particularly valuable for cancer modeling, investigating tumor–immune cell interaction, and providing diagnostic, predictive, and therapeutic value to boost cancer therapy.

*Future Directions*—Despite the success of immunotherapy in many types of cancer, only 20–30% of tumor patients across tumor types have been shown to be able to benefit from ICB treatment. Even in those small portions of patients with response, some of them will acquire resistance to their treatment. Thus, one of the future directions is to identify robust biomarkers predictive of response (or resistance). Another direction is to leverage these technologies to study mechanisms of resistance and further identify and validate new therapeutic regimen(s) to overcome resistance. One potential approach is to integrate microfluidic technology with other various technologies. For example, integration of scRNA-seq and microfluidic technology allows profiling immune cells in devices, which have been shown to predict response to anti-PD1 therapy [202]. Alternatively, performing BH3 profiling to identify apoptotic blocks in cancer cells has been shown to provide diagnostic value to guide effective rational therapies [203]. In addition, isolating resistant cells after treatment presents an opportunity to study resistance in detail and ultimately find a way to circumvent it. While researchers have proposed many strategies to overcome resistance, future developments should focus on providing approaches to evaluate their efficacy in reliable preclinical murine tumor models [60]. This will not only reduce the cost of animal testing, but also improve the rate of success in clinical trials. Furthermore, these devices must be able to conduct multiple different tests in a singular chip to fit their high-throughput features. On the other hand, developing next-generation microfluidic models which can model tumor-draining lymph nodes, recruitment of naïve lymphocytes, and TLS formation/activity represents another direction to advance our understanding of TME, promote drug development, and overcome drug resistance.

As microfluidic devices have been reported to be used in diagnosis, response prediction, response monitoring, resistance study, and drug efficacy validation, one obvious question is whether 3D microfluidic models can/will become clinical tools for tailored, personalized, and precision cancer therapies. Recently, Ooft et al. conducted a clinical trial based on patient-derived organoid drug response, providing a cautionary tale [115]. In this study, 31 organoids were generated from 54 eligible patients out of 61, and 25 of them were subjected to drug screening, with 19 organoids showing responses to one or more drugs. However, despite drug sensitivity in organoids, patients did not demonstrate clinical responses with the same treatment. This suggested that parameters, including culture success rate, clinical deterioration of patients during standard of care, and rational design of drug panels, should be considered in organoid-guided clinical studies and probably other similar models. Although there are issues in current models before they can be applied into clinical studies, some microfluidic chips have been approved by the FDA, such as the CellSearch Chip being used for CTC enrichment [168]. Efforts are needed from many aspects to address the above issues to evolve microfluidic devices from bench to bedside.

## 7. Conclusions

Cancer therapies have advanced substantially over the past decade, especially the use of targeted therapies and immunotherapies. Despite the success of both targeted and immune-based cancer therapies, intrinsic and acquired resistance remains a persistent challenge. Advancing our understanding of the heterogeneity of the tumor and TME and the dynamic nature of tumor–immune interactions will require further investigation. With multiple combination therapies, new biomarkers, and inhibitory receptors being proposed to guide our approaches to diagnosis, prognosis, and therapy, there is a need for more sophisticated preclinical models that translate to human immunity and provide reliable functional applications, such as studying organ-specific immune contexture and allowing efficient/effective assessment of immunotherapy combinations. The use of microfluidic devices maintaining physiological accuracy and features in patient tumors and the TME for studies in cancer immunotherapy provides new opportunities. The integration of 3D culture with microfluidic technology to model tumor–immune dynamics, whether using explanted patient-derived tumor tissue or “enhanced” models in which immune cells are added, holds promise to address the remaining challenges. The estimated cost for a successful cancer drug is about 1 billion USD, and roughly 90% of drugs entering phase I clinical trials fail to offer clinical benefit and are not developed further. Thus, leveraging microfluidic technology with 3D culture systems, ideally with patient-derived tumor tissue, to evaluate efficacy has the potential to reduce time and cost for drug development. Furthermore, microfluidic devices are amendable to perform several multi-omics studies and real-time imaging analyses, which will enable the study of tumor–immune interactions and mechanisms of therapy resistance, as well as the identification and evaluation of novel cancer therapeutics. Taken together, it is promising that microfluidic modeling of the tumor microenvironment can potentially be incorporated into clinical practice to advance cancer immunotherapy and precision medicine.

## Figures and Tables

**Figure 1 cancers-13-06052-f001:**
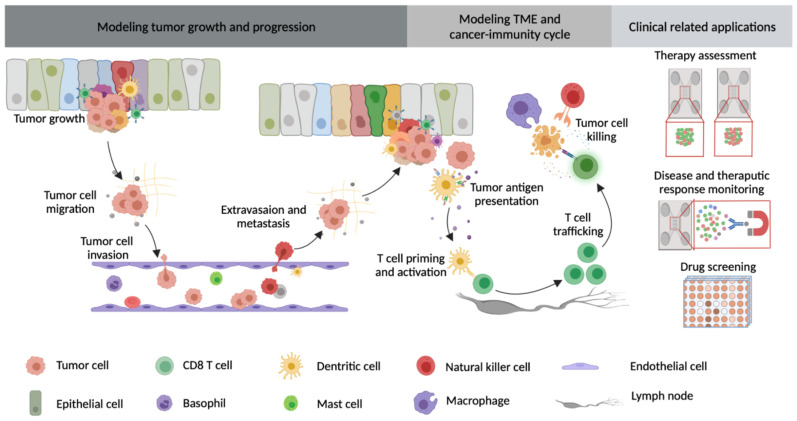
Microfluidic devices in modeling tumor growth and progression, TME, and cancer–immunity cycle, as well as clinical-related applications.

**Table 1 cancers-13-06052-t001:** TME models.

Type	Models	Material Source	Applications	Advantages	Disadvantages	Reference
In vivo murine models	Syngeneic tumor models	-Immune-competent mice: C57BL/6, BALB/c, FVB, etc. -Transplantable cells: B16, 4T1, CT26, etc.	-Tumor formation and progression -Evaluate antitumor immune response	-Have physiologically relevant tumor microenvironment -Easy to manipulate	-Variability of phenotype because of the site of engraftment -Lack of heterogeneity	[43,44,45,46,47]
	Genetically engineered mouse models (GEMM)	-Immune-competent mice: C57BL/6, etc.	-Autochthonous tumor development -Evaluate antitumor immune response -Modeling immune-related adverse events (irAEs)	-Have naïve TME -Tumor initiation and progression driven by relevant genetic alterations	-Variability in tumor penetrance and latency -Low immunogenicity due to defined alterations	[48,49,50]
	Humanized mouse	-Immune-deficient mice: SCID, NOD, NSG, etc.	-Evaluate antitumor therapies	-Reproduce genomic heterogeneity of human disease -Have reconstituted human immune system	-Require autologous immune system reconstitution -Low rates and duration of immune reconstitution	[45,51,52,53]
2D	Coculture	-Tumor cells -TME components (macrophages, dendritic cells, fibroblast cells, etc.)	-Study the interaction between tumor and immune cells (cytokine secretion, tumor killing, etc.)	-Easy to manipulate -Can be used in high-throughput study	-Lack of native immune and stromal components -Limited reflection in tumor morphological phenotype	[54,55]
3D	Spheroids	Coculture: cell lines, mouse- or patient-derived tissues, and other TME components (macrophages, T cells, etc.)	-Study the interaction between tumor and immune cells -Evaluate antitumor immune response	-Easy to manipulate -Can reflect genetic alterations and keep morphological phenotype of original tumor	-Lack of native immune and stromal components	[56,57]
		Microfluidic devices: cell lines, mouse- or patient-derived tissues	-Study the interaction between tumor and immune cells -Evaluate the efficacy of therapeutic combinations -Profile secreted cytokines	-Require limited material (cells, media, reagents, etc.) -Can reflect genetic alterations and keep morphological phenotype of original tumor -Preserve immune cell population in TME	-Size limitation -Require microfluidic devices -Only have native tumor-infiltrating immune cells -Cannot model T-cell trafficking	[58,59,60]
	Organoids	Coculture: mouse- or patient-derived tissues and other TME components (macrophages, dendritic cells, etc.)	-Evaluate antitumor immune response -Assessment of tumor organoid killing	-Easy to enrich and expand tumor organoids -Can reflect genetic alterations and keep morphological phenotype of original tumor	-Lack of native immune and stromal components	[61,62,63]
		ALI (Air-Liquid Interface) culture: mouse or patient-derived tissues	-Study the interaction between tumor and immune cells -Evaluate antitumor immune response -Assessment of tumor organoid killing	-Can reflect genetic alterations and keep morphological phenotype of original tumor -Preserve multiple immune cells and fibroblasts in TME	-Only have native tumor-infiltrating immune cells -Cannot model T-cell trafficking	[61,62,64]

**Table 2 cancers-13-06052-t002:** Microfluidic technology in cancer modeling.

Applications	Models	Experiment Design	Microfluidic Features	Reference
Cancer growth and progression	Tumor growth	Coculture cancer cells with fibroblasts	With fibronectin-rich matrix	[99]
		Culture cancer cells with/without treatment of fibrin	A bifurcated microfluidic device allowing comparison between two different cell environments	[148]
	Tumor migration and extravasation	Treat cancer cells with different secreted factors	Use a monolayer of endothelial cells to mimic microvasculature	[149,150]
		Coculture cancer cells with fibroblast		[151]
	Angiogenesis	Test the effects of multiple angiogenic factors on angiogenesis	Use biomimetic model to reconstitute angiogenic sprouting in microfluidic device	[152,153]
	Cancer metastasis	Treat cancer cells with proinflammatory cytokine (e.g., IL-6)	Have lymph vessel–tissue–blood vessel structure	[154]
		Treat cancer cells with proinflammatory cytokine (e.g., TNFα)	Use a monolayer of endothelial cells to mimic microvasculature	[155]
TME and cancer–immunity cycle	TME modeling	Coculture tumor spheroids with other TME components (e.g., CAF, stroma cells, endothelial cells)	Culture spheroids	[156,157,158]
		Vascularized system modeling	Contain interconnected microchannels to model a highly vascularized system	[159]
		ECM and interstitial flow modeling	Modeling biophysical features, such as ECM and interstitial flow in TME	[160]
		Oxygen concentration modeling	Include three parallel connected tissue chambers and an oxygen scavenger channel to control oxygen concentration	[161,162]
		Study the interaction between tumor and immune cells	Culture murine- and patient-derived organotypic tumor spheroids (MDOTSs/PDOTSs)	[60,138]
	Immune cell migration/recruitment	Identify potential factors (e.g., chemokine, cytokines) affecting immune cell migration	Integrate microscopy technology with microfluidic chips or use microfluidic devices designed for co-culture	[98,103,163,164]
	T lymphocyte activation	Monitor T-cell activation by analyzing CD69 expression	Use a chip containing microelectrodes to get dielectrophoretic manipulation	[165]
		Monitor T-cell activation by analyzing the binding of T cells to TNFα-treated human umbilical vein endothelial cells (HUVECs)	Adjustable shear stress	[166]
Clinica- related applications	Therapy assessment	Evaluate the efficacy of therapeutic combinations	Culture MDOTSs/PDOTSs	[60,138]
		Coculture cancer spheroids with natural killer cells or antibody–cytokine regimens	Use a monolayer of endothelial cells to mimic microvasculature	[167]
	Disease and therapeutic response monitoring	Analyze CTCs from patients to predict prognosis and evaluate progression-free survival and overall survival of patients	Exploit antibody-coated magnetic particles targeting EpCAM to detects and quantify CTCs	[168,169,170,171,172]
		Capture exosome to monitor immunotherapeutic response	Employ immunomagnetic beads/antibodies/chips to capture and measure exosomal tumor markers	[173,174,175]
		Study intra-tumoral heterogeneity in microfluidic devices with scRNA-seq and understand therapeutic evasion	Incorporate different scRNA-seq techniques into microfluidic chips (e.g., droplet microfluidics, Microwell-seq microfluidics)	[176,177,178]
		Monitor immune cell heterogeneity	Timelapse imaging microscopy-based microfluidic platform	[179]
			Microfluidic devices integrating single- cell barcoding chip (SCBC) or antibody microarray (BOBarray)	[180,181,182]
	Drug screening	Test drug toxicity with bio-printed hepatic spheroids	Use hepatic spheroids as material source to directly print liver tissue into the microfluidic device	[183]
		Evaluate the response of thyroid tissue to radioiodine sensitivity/adjuvant therapies in real time	Culture live-sliced human thyroid tissue	[184]
		Provide dynamic and combinatorial drug screening	Culture pancreatic organoids	[185]
		Chemotherapeutic drug testing and efficacy evaluation	Integration of microfluidics and electrical sensing modality	[186]

This table summarizes the applications of microfluidic devices in different cancer models. CTC, circulating tumor cell; EpCAM, epithelial cellular adhesion molecule; scRNA-seq, single-cell RNA sequencing; CAF, cancer-associated fibroblast.

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
