# Peer review of "Going with the Flow: Modeling the Tumor Microenvironment Using Microfluidic Technology"

_cancers, 2021, doi:10.3390/cancers13236052_

Round 1

Reviewer 1 Report

Xie et al present an overview of the use of microfluidic technology to model the tumor microenvironment. This is a rapidly evolving field, and the authors have put together a highly relevant review. Apart from a few minor issues, this is an interesting synopsis of current research.

Minor issues:

1) Spelling and correct grammar should be checked. Particularly from chapter 5 on, these mistakes are slightly disturbing. Also, some sentence are very long and difficult to read.

2) Some abbreviations (e.g. ICI, TLS) are not explained; please do so.

3) Some general thought on how close microfluids technology mimics the tumor situation in vivo  should be added (in Chapter 4 or Chapter 6 - here: How could this be improved generally?)

4) Table 2 should have a legend that better describes its content. Only spheroids were used in all of the studies? Organoids are not mentioned at all.

Author Response

1) Spelling and correct grammar should be checked. Particularly from chapter 5 on, these mistakes are slightly disturbing. Also, some sentence are very long and difficult to read.

Thank you for the careful review of our manuscript.  We have revised manuscript in several places to improve grammar and correct spelling mistakes.

2) Some abbreviations (e.g. ICI, TLS) are not explained; please do so.

TLS (tertiary lymphoid structure) has been indicated in the next. ICI was removed and replaced with "ICB" (immune checkpoint blockade), which is used throughout this text.

3) Some general thought on how close microfluids technology mimics the tumor situation in vivo  should be added (in Chapter 4 or Chapter 6 - here: How could this be improved generally?)

A brief section has been added in Ch. 6 ("challenges") that addresses the challenges of trying to mimic the tumor microenvironment using microfluidic devices.

4) Table 2 should have a legend that better describes its content. Only spheroids were used in all of the studies? Organoids are not mentioned at all.

We have added a brief legend and added details regarding use of organoids, etc. for specific sections.

Reviewer 2 Report

This paper is well written and helpful for researchers on tissue engineering. TME has been recently noted in cancer research because TME affects cancer diseases. For drug screening, 3D models, in vivo models, and the microfluidic device is essential. However, the important concept is a lack in the paper. Biomaterials are crucial in designing 3D models. The description is needed for readers’ better understanding. The paper would be accepted only when the below comments are responded.

1.

3.3.2.

To construct the 3D cell culture model, biomaterials enable the interaction between cancer cells and stromal cells. The microfluidic device is one of the materials, so the authors should describe biomaterial and 3D TME models. At least recent papers should be added for the additional sentences. I suggest these papers be added.

Review papers (for the overview description of concept)

Cancers 202012(10), 2754.

Tissue Engineering Part B: 2010. 351-359.

Research papers

Tissue Eng. Part C Methods 201925, 711–720.

Biomaterials 77 (2016) 164-172.

ACS Biomater. Sci. Eng. 2020, 6, 1, 539–552.

Tissue Eng. Part A, 26, 2020, 1272-1282.

PNAS 2013 110 (16) 6500-6505;

2.

Information about various types of cancers should also be included in this review paper.

Author Response

1. regarding 3.3.2. To construct the 3D cell culture model, biomaterials enable the interaction between cancer cells and stromal cells. The microfluidic device is one of the materials, so the authors should describe biomaterial and 3D TME models. At least recent papers should be added for the additional sentences. I suggest these papers be added.

Review papers (for the overview description of concept)

Cancers 202012(10), 2754.

Tissue Engineering Part B: 2010. 351-359.

Research papers

Tissue Eng. Part C Methods 201925, 711–720.

Biomaterials 77 (2016) 164-172.

ACS Biomater. Sci. Eng. 2020, 6, 1, 539–552.

Tissue Eng. Part A, 26, 2020, 1272-1282.

PNAS 2013 110 (16) 6500-6505;

we thank the reviewer for this important criticism and have added brief discussion on the importance of selection of biomaterials in section 3.3.2

2. Information about various types of cancers should also be included in this review paper.

While specific examples of use cases involving different cancer types are mentioned throughout the manuscript, we agree that that this is not made clear in a single statement. We have included a statement in section 3.3.2 (after the discussion of biomaterials) that details the use of different cancer models in 3D.

Round 2

Reviewer 2 Report

The authors have responded to all comments, and the papers have been significantly improved. Therefore, I recommend the acceptance for publication.

In proofreading, please combine and reflect the uncategorized references to reference lists.